# Specialty Care and Counselling about Hereditary Cancer Risk Improves Adherence to Cancer Screening and Prevention in Newfoundland and Labrador Patients with *BRCA1/2* Pathogenic Variants: A Population-Based Retrospective Cohort Study

Aimee Roebothan [1], Kerri N. Smith [2,3], Melanie Seal [4], Holly Etchegary [5] and Lesa Dawson [6,*]

1 Faculty of Medicine, Memorial University, St. John's, NL 1AB 3V6, Canada; e59amh@mun.ca
2 Centre for Translational Genomics, NL Health Services, St. John's, NL 1AB 3V6, Canada
3 Discipline of Laboratory Medicine, Faculty of Medicine, Memorial University, St. John's, NL 1AB 3V6, Canada
4 Discipline of Oncology, Faculty of Medicine, Memorial University, St. John's, NL 1AB 3V6, Canada; melanie.seal@easternhealth.ca
5 Community Health and Humanities, Faculty of Medicine, Memorial University, St. John's, NL 1AB 3V6, Canada; holly.etchegary@med.mun.ca
6 Division of Gynecologic Oncology, Faculty of Medicine, Memorial University, St. John's, NL 1AB 3V6, Canada
* Correspondence: lmdawson@mun.ca

**Abstract:** Pathogenic variants (PVs) in *BRCA1* and *BRCA2* increase the lifetime risks of breast and ovarian cancer. Guidelines recommend breast screening (magnetic resonance imaging (MRI) and mammogram) or risk-reducing mastectomy (RRM) and salpingo-oophorectomy (RRSO). We sought to (1) characterize the population of *BRCA1/2* PV carriers in Newfoundland and Labrador (NL), (2) evaluate risk-reducing interventions, and (3) identify factors influencing screening and prevention adherence. We conducted a retrospective study from a population-based provincial cohort of *BRCA1/2* PV carriers. The eligibility criteria for risk-reducing interventions were defined for each case and patients were categorized based on their level of adherence with recommendations. Chi-squared and regression analyses were used to determine which factors influenced uptake and level of adherence. A total of 276 *BRCA1/2* PV carriers were identified; 156 living NL biological females composed the study population. Unaffected females were younger at testing than those with a cancer diagnosis (44.4 years versus 51.7 years; $p = 0.002$). Categorized by eligibility, 61.0%, 61.6%, 39.0%, and 75.7% of patients underwent MRI, mammogram, RRM, and RRSO, respectively. Individuals with breast cancer were more likely to have RRM (64.7% versus 35.3%; $p < 0.001$), and those who attended a specialty hereditary cancer clinic were more likely to be adherent to recommendations (73.2% versus 13.4%; $p < 0.001$) and to undergo RRSO (84.1% versus 15.9%; $p < 0.001$). Nearly 40% of the female *BRCA1/2* PV carriers were not receiving breast surveillance according to evidence-based recommendations. Cancer risk reduction and uptake of breast imaging and prophylactic surgeries are significantly higher in patients who receive dedicated specialty care. Organized hereditary cancer prevention programs will be a valuable component of Canadian healthcare systems and have the potential to reduce the burden of disease countrywide.

**Keywords:** BRCA; breast cancer; ovarian cancer; screening prevention; health policy genetics

## 1. Introduction

The *BRCA1* and *BRCA2* genes encode proteins responsible for double-strand DNA repair [1]. Pathogenic variants (PV) in these genes cause a hereditary cancer predisposition syndrome resulting in lifetime risks of breast cancer (BC) and ovarian cancer (OC) of 51–72% and 11–44%, respectively, a significant contrast when compared to rates of 13% and 1.3% in

the general population [1–4]. Since the identification of these genes in 1994 and 1995 [5,6], evidence has grown to support the development of clear recommendations regarding the optimal management of *BRCA1/2* PV carriers, through which cancer rates and all-cause mortality can be improved [7–9].

Many cancers arising in females with a *BRCA1/2* PV can be prevented. Canadian guidelines recommend annual breast magnetic resonance imaging (MRI) starting at age 25 and an annual mammogram starting at age 30 for females with a *BRCA1/2* PV [10]. These surveillance recommendations differ from those for females with population risk, which recommend mammograms starting at age 50 every two to three years [11]. Annual breast MRI is associated with decreased BC stage, lower overall progression to metastatic disease, and increased survival [12–15], and confers a sensitivity of >90% for early-stage BC detection when combined with mammogram [16]. Risk-reducing mastectomy (RRM) nearly eliminates the BC risk in female *BRCA1/2* PV carriers and should be discussed with all patients, with thoughtful consideration of the personal nature of this decision for affected individuals [17]. Given the high mortality and lack of effective OC screening, risk-reducing salpingo-oophorectomy (RRSO) is the only effective OC prevention, recommended between ages 35 to 40 and 40 to 45 in *BRCA1* and *BRCA2* PV carriers, respectively [10]. This procedure is typically quite safe, with a low surgical complication rate, and most patients will have an outpatient, minimally invasive laparoscopic procedure [18]. The resulting premature menopause can be managed safely with hormone replacement therapy (HRT) in patients without a history of estrogen-sensitive BC. For all patients with premature surgical menopause, preventative care to address the increased risk of osteoporosis and cardiovascular disease is essential [19]. In patients where HRT is contraindicated, vasomotor symptoms can be managed very effectively with non-hormonal interventions [20]. RRSO decreases the risk of OC and BC by >80% and 50%, respectively, with a 77% reduction in all-cause mortality and is the cornerstone of effective cancer prevention in these high-risk individuals [21,22]. BC risk may also be reduced after RRSO in *BRCA2* PV carriers [23]. In addition to the improved overall health and survival outcomes, surveillance and risk-reducing surgeries are cost effective for healthcare systems [24–26].

Despite clear evidence that these interventions are effective, many Canadian jurisdictions have yet to implement programmatic follow-up of high-risk patients. After initial genetic counselling and result disclosure, navigation of and compliance with recommendations is often left solely to the individual and their primary care physician. In an era of an aging Canadian population, increasingly complex cancer treatments, improving survivorship, and the debut of costly targeted therapeutics, a cancer care system focused on treatment over prevention will not be sustainable.

Considering that many factors may influence a specific patient's ability to access recommended care, it will be important to better understand barriers and predictors of intervention uptake. It is not known, for example, if geographic distance to healthcare centers, patient age, prior cancer diagnosis, access to specialty cancer genetics care, or family history of cancer might each influence how likely it is that a given patient will make use of screening or prevention. Future programs offering support and navigation can be better designed if the influence of these factors can be understood.

The Newfoundland and Labrador (NL) population of 510,550 [27] individuals includes Indigenous peoples and those of English, Irish, and French ancestry. The well-known NL founder population [28], as well as its population-based healthcare system, centralized cancer care, and medical genetics programs, and province-wide electronic medical record, makes this province an ideal location for genetics and health service delivery research. In this context, we sought to: (1) characterize the population-based cohort of female NL *BRCA1/2* PV carriers, (2) evaluate the uptake of risk-reducing interventions, and (3) identify factors that influence the uptake of screening and prevention.

## 2. Materials and Methods

### 2.1. Inclusion and Exclusion Criteria

Records from the Provincial Medical Genetics Program, Eastern Health Authority, St. John's, NL, Canada were queried to obtain a complete population-based dataset of all *BRCA1/2* PV carriers identified through both clinical (2006–2017) and research (1994–2006) programs. Records from the gynecologic oncology Inherited Cancer Prevention Clinic (ICPC) also included individuals with a *BRCA1/2* PV identified via private or out-of-province testing. A retrospective review of the electronic medical records of all NL female *BRCA1/2* PV carriers who were at least 18 years of age collected demographics, genetic testing reports, and pathology, as well as the uptake of risk-reducing interventions including MRI, mammography, RRM, and RRSO. The cases included in the analyses were those with biologic female sex assigned at birth, *BRCA1* or *BRCA2* PV carriers who were at least 18 years of age, alive, and living in NL. Excluded from the study were males, and females not currently living in the province or who were deceased at the time of data collection.

### 2.2. Design

The eligibility criteria for each risk-reducing intervention were strictly defined according to recommendations [10,29–31]. The eligibility criteria for MRI and mammogram screening were those women with breast(s) at the time of data analysis and who were 25 to 75 years of age (MRI) and 30 to 75 years of age (mammogram). *BRCA1/2* PV carriers receiving cancer treatment in the last 18 months were excluded, as they would typically not undergo screening. Breast screening was considered adequate when undertaken within 18 months of the analysis. This timeframe was selected despite the 12-month recommendation to allow "real-world" latitude around scheduling within menstrual cycles and appointment availability. Patients eligible for RRM included those women with breast(s) at the time of genetic testing and who were between the ages of 25 and 75 years of age at any time from genetic testing until data analysis. The RRSO eligibility criteria were defined by the specific *BRCA1/2* PV and age; women with ovaries at the time of genetic testing were considered eligible for RRSO if they were between 35 and 75 years of age with a *BRCA1* PV and between 40 and 75 years of age with a *BRCA2* PV at any time from genetic testing until data analysis. Women with metastatic BC whose cancer therapy did not include therapeutic oophorectomy were not considered eligible for RRSO.

The population was categorized into one of three groups based on their level of screening and prevention adherence. Females with high adherence were those between 25 and 75 years of age who were both adherent to MRI breast screening or RRM, and had completed RRSO (if eligible). Adherent females were those between 25 and 75 years of age who were adherent to either MRI breast screening or RRM, or RRSO (if eligible). Non-adherent females included those between 25 and 75 years of age with no MRI breast screening, RRM, or RRSO (if eligible).

### 2.3. Statistical Analysis

The proportions of uptake and adherence to each intervention were assessed. Uptake predictors were explored by comparing potential factors influencing compliance, including geographic distance to a healthcare center, prior cancer diagnosis, diagnosis of BC/OC in relatives, and specialist assessment by a medical oncologist and/or the ICPC. Analyses were performed using unpaired t-tests, chi-squared tests (uptake), and multinomial logistic regression (adherence) using SPSS Statistics version 27 (IBM). $p < 0.05$ was considered statistically significant.

## 3. Results

### 3.1. Study Population

Figure 1 outlines the study inclusion criteria and assigns the categories of those eligible for each screening and prevention intervention according to individual cases and relevant recommendations [10,29–31]. A description of the population is shown in Table 1. Of

156 living NL female *BRCA1/2* carriers, 57 (36.5%) had a *BRCA1* PV and 99 (63.5%) had a *BRCA2* PV (*p* < 0.001). A total of 99 (63.5%) females were unaffected and underwent genetic testing because of a known familial *BRCA1* or *BRCA2* PV, whereas 57 (36.5%) had a cancer diagnosis; 51 (32.7%) had BC and 8 (5.1%) had OC. Of all females with cancer, 49 (31.4%) were diagnosed before genetic testing (BC: *n* = 44; OC: *n* = 5) and 8 (5.1%) were diagnosed after genetic testing (BC: *n* = 5; OC: *n* = 3). Unaffected females were younger at the time of testing than those with a personal cancer history (44.4 years versus 51.7 years; *p* = 0.002). *BRCA1* PV carriers had higher BC rates compared to those with a *BRCA2* PV (40.4% versus 28.3%), and were also a younger age at BC (43.8 years versus 48.5 years) and OC (47.5 years versus 55.8 years) diagnosis, although neither metric reached statistical significance. *BRCA1* PV carriers were significantly younger when they completed genetic testing (43.7 years versus 49.1 years; *p* = 0.020). Significantly more *BRCA2* PV carriers availed of specialty care from a medical oncologist (45.5% versus 62.1%; *p* = 0.048) or a medical oncologist and/or the ICPC (67.3% versus 83.2%; *p* = 0.025).

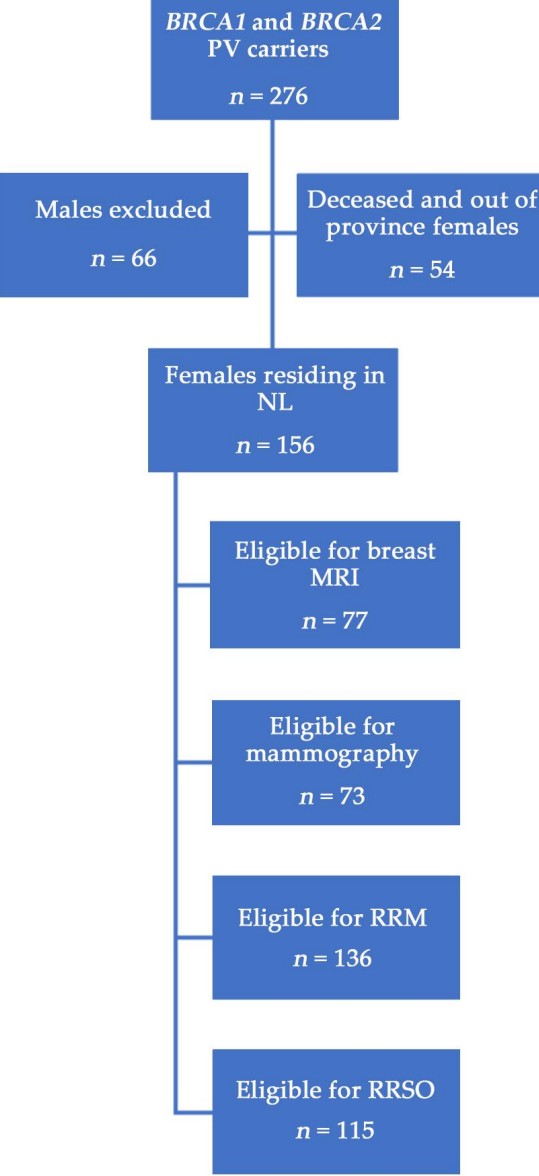

**Figure 1.** Study population of *BRCA1/2* PV carriers in NL. Eligibility for surgeries or screening was based on age and individual case, e.g., females with prior bilateral RRM were ineligible for breast screening, patients under 30 years of age were not eligible for mammography, etc.

**Table 1.** Demographic and clinical characteristics of female *BRCA1/2* PV carriers in NL.

| Characteristic | Overall | *BRCA1* PV | *BRCA2* PV | *p*-Value * |
|---|---|---|---|---|
| Total (*n*) | 156 | 36.5% (57) | 63.5% (99) | **<0.001** |
| Age at time of study (years) | | | | |
| Mean (± SD) | 53.3 ± 14.2 | 52.1 ± 13.6 | 54.0 ± 14.5 | 0.412 |
| Median | 54.0 | 54.0 | 53.0 | |
| Range | 24–89 | 24–79 | 28–89 | |
| Age at genetic testing (years) | | | | |
| Mean (± SD) | 47.1 ± 14.0 | 43.7 ± 12.7 | 49.1 ± 14.4 | **0.020** |
| Median | 45.5 | 42.0 | 50.0 | |
| Range | 18–83 | 18–67 | 24–83 | |
| Genetic testing after BC/OC diagnosis | 31.4% (49) | 79.2% (19) | 96.8% (30) | **0.038** |
| BC and/or OC (*n*) | 36.5% (57) | 45.6% (26) | 31.3% (31) | 0.074 |
| BC (*n*) | 32.7% (51) | 40.4% (23) | 28.3% (28) | 0.122 |
| Age at BC diagnosis (years) | | | | |
| Mean (± SD) | 46.5 ± 9.7 | 43.8 ± 9.5 | 48.5 ± 9.5 | 0.091 |
| Median | 44.0 | 42.0 | 49.0 | |
| Range | 31–65 | 31–64 | 32–65 | |
| OC (*n*) | 5.1% (8) | 7.0% (4) | 4.0% (4) | 0.417 |
| Age at OC diagnosis (years) | | | | |
| Mean (± SD) | 51.6 ± 11.4 | 47.5 ± 13.0 | 55.8 ± 9.5 | 0.344 |
| Median | 52.0 | 47.5 | 56.5 | |
| Range | 32–66 | 32–63 | 44–66 | |
| Assessed by medical oncology (*n*) | 56.0% (84) | 45.5% (25) | 62.1% (59) | **0.048** |
| Assessed by ICPC (*n*) | 56.7% (85) | 49.1% (27) | 61.1% (58) | 0.154 |
| Received specialty care † (*n*) | 77.3% (116) | 67.3% (37) | 83.2% (79) | **0.025** |
| Primary care provider on record (*n*) | 95.7% (135) | 96.3% (52) | 95.4% (83) | 0.798 |
| Rural residence ‡ (*n*) | 39.1% (61) | 35.1% (20) | 41.4% (41) | 0.436 |

\* Comparison of *BRCA1* and *BRCA2* PV carriers; significant *p*-values (<0.05) are shown in bold font. † Includes those women who have been assessed by medical oncology and/or the ICPC. ‡ Rural areas are those with postal codes containing zero as the first number (e.g., A0A 1A3).

The specific *BRCA1/2* PVs observed are outlined in Figure 2. A total of 18 unique *BRCA1* PVs were observed, with frequencies ranging from 1.8% (*n* = 1) to 36.8% (*n* = 21) within the *BRCA1* PV carrier population. Two *BRCA1* PVs (c.2071delA and c.2999delA) had frequencies over 10%. Similarly, 20 unique *BRCA2* PVs were identified; frequencies ranged from 1.0% (*n* = 1) to 19.2% (*n* = 19) amongst all *BRCA2* PV carriers. Three *BRCA2* PVs (c.7988A>T, c.4876_4877delAA, and c.6065C>G) were identified at a frequency higher than 10%. The pathologic stage, cell type, and receptor status for the cancer diagnoses is shown in Table 2. Among those cases with available pathology reports, the most common BC tumor histology was invasive ductal carcinoma, and all OC cases were serous. The majority of BC cases were diagnosed at stage II (*n* = 18), and the most common receptor status was triple negative (*n* = 12).

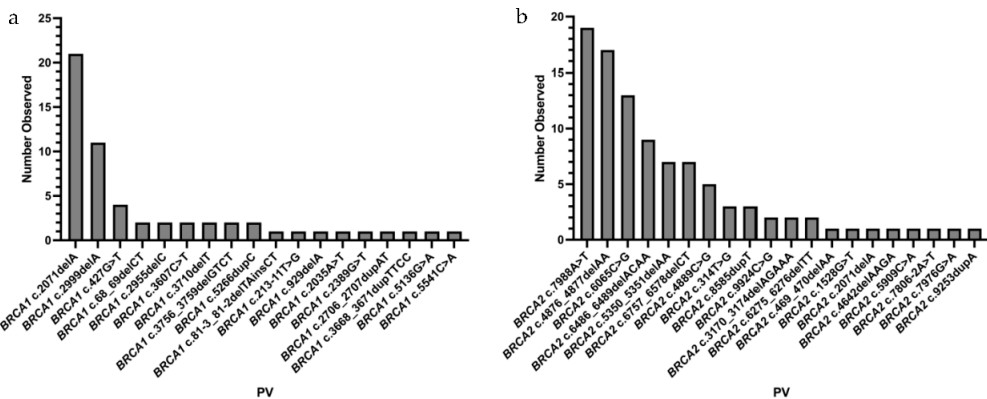

**Figure 2.** *BRCA1* (**a**) and *BRCA2* (**b**) PVs identified in NL females.

**Table 2.** Clinical characteristics of female NL *BRCA1/2* PV carriers diagnosed with BC and/or OC.

| Patient * | Tumor Type | Age of Onset | Tumor Histology | Stage | Grade | Receptor Status † |
|---|---|---|---|---|---|---|
| 1 | Breast | | | | | |
| 2 | Breast | 49 | IDC | II | | ER+PR+Her2- |
| 3 | Breast | 33 | IDC | II | | ER-PR-Her2- |
| 4 | Breast | 42 | DCIS | II | | ER-PR- |
| 5 | Breast | 37 | IDC | II | 3 | ER-PR-Her2- |
| 6 | Breast | 34 | IDC | | 3 | ER+PR-Her2- |
| 7 | Breast | 40 | IDC | | 3 | |
| 8 | Breast | 34 | | II | | ER-PR- |
| 9 | Ovary | 63 | Serous | II | 3 | n/a |
| 10 | Breast | 50 | | | | ER-PR-Her2- |
| 11 | Breast | 48 | IDC | II | 3 | ER-PR-Her2- |
| 12 | Breast | 31 | IDC | | | |
| 13 | Breast | 50 | IDC | I | 3 | ER-PR- |
| 14 | Breast | 42 | IDC | II | 3 | ER+PR+Her2- |
| 15 | Breast | 34 | IDC | II | 3 | ER+PR+Her2+ |
| 16 | Breast | 37 | IDC | I | 3 | ER-PR-Her2- |
| 17 | Breast | 41 | MDC | I | | ER-PR+ |
| 18 | Breast | 65 | IDC | | | |
| 19 | Breast | 64 | IDC | II | 3 | ER-PR-Her2- |
| 20 | Breast | 41 | DCIS | I | | |
| 21 | Breast | 33 | IDC | | | |
| 22 | Breast | 42 | DCIS | | | |
| 23 | Breast | 41 | IDC | II | 3 | ER-PR- |
| 24 | Breast | 37 | IDC | | | |
| 25 | Breast | 64 | IDC | I | 3 | ER-PR-Her2- |
| 26 | Breast | 51 | IDC | II | 3 | ER+PR-Her2- |
| 27 | Breast | 56 | IDC | I | 1 | ER+PR+Her2- |
| 28 | Breast | | IDC | | | |
| 29 | Breast | 44 | IDC | II | 3 | ER-PR- |
| 30 | Breast | 40 | IDC | I | | ER-PR- |
| | Ovary | 60 | | | | n/a |
| 31 | Breast | 61 | IDC | | 2 | ER+PR+Her2- |
| 32 | Breast | 57 | IDC | III | 3 | ER-PR-Her2- |
| 33 | Breast | 53 | DCIS | | 3 | |
| 34 | Breast | 51 | IDC | II | 3 | ER-PR-Her2+ |
| 35 | Breast | 36 | IDC | II | 3 | ER-PR- |
| 36 | Breast | 43 | IDC | II | 3 | ER-PR-Her2- |
| 37 | Breast | 44 | IDC | I | 3 | ER-PR-Her2- |
| 38 | Breast | 45 | IDC | II | 2 | ER+PR+Her2- |
| 39 | Breast | 42 | DCIS | 0 | 2 | ER+PR+Her2+ |
| 40 | Ovary | 53 | Serous | | | n/a |
| 41 | Breast | 60 | IDC | II | 3 | ER+PR+Her2- |
| | Ovary | 51 | | | | n/a |
| 42 | Breast | 55 | IDC | I | 3 | ER-PR-Her2- |
| 43 | Breast | 41 | IDC | | | |
| 44 | Breast | 59 | DCIS | | | |
| 45 | Breast | 43 | IDC | II | 3 | ER+PR+Her2+ |
| 46 | Breast | 32 | IDC | | 3 | |
| 47 | Breast | 53 | IDC | I | 3 | ER-PR-Her2- |
| 48 | Breast | 41 | DCIS | | | |
| 49 | Breast | 63 | IDC | I | 2 | ER+PR-Her2- |
| 50 | Ovary | 66 | Serous | III | 3 | n/a |
| 51 | Ovary | 44 | Serous | III | 2 | n/a |
| 52 | Breast | 57 | IDC | | 3 | ER+PR+ |
| 53 | Breast | 49 | DCIS | | 2 | |
| 54 | Breast | 56 | IDC | III | 3 | ER+PR-Her2- |
| 55 | Breast | 56 | DCIS | I | 3 | |
| 56 | Ovary | 32 | Serous | IV | | n/a |
| 57 | Ovary | 44 | Serous | II | | n/a |

DCIS: ductal carcinoma in situ; IDC: invasive ductal carcinoma; MDC: medullary ductal carcinoma; n/a: not applicable. * Missing data has been left blank. † Receptors for which data are missing are excluded.

### 3.2. Screening and Preventative Interventions

The screening and risk-reducing intervention uptake are shown in Table 3 and Figure 3. Categorized by individual eligibility, significant proportions of females underwent MRI (61.0%), mammogram (61.6%), or RRSO (75.7%) (all $p < 0.001$); a significant minority

underwent RRM (39.0%; $p$ = 0.025). There were no significant differences in intervention uptake between *BRCA1* versus *BRCA2* PV carriers when classified by age and eligibility. On univariate analysis, consultation with specialty care, as provided by the ICPC and/or a medical oncologist, was strongly correlated with RRSO uptake (84.1% versus 15.9%; $p < 0.001$). Females with a personal history of BC were significantly more likely to complete RRM compared to those without a prior BC diagnosis (64.7% versus 35.3%; $p < 0.001$). No differences were observed in individual intervention uptake amongst those who lived remotely from care centers, those without a family physician, or those aged 50 or older.

**Table 3.** Chi-squared analysis of factors influencing screening and risk-reducing intervention uptake among eligible female NL *BRCA1/2* PV carriers.

| | **Risk-Reducing Intervention *** | | | | | | | |
|---|---|---|---|---|---|---|---|---|
| **Eligible Uptake** $p$**-Value †** | **MRI** 49.4% (77) 61.0% (47) <0.001 | | **Mammogram** 46.8% (73) 61.6% (45) <0.001 | | **RRM** 88.3% (136) 39.0% (53) 0.025 | | **RRSO** 67.3% (115) 75.7% (87) <0.001 | |
| **Factor** | **% (*n*)** | ***p*-Value** | **% (*n*)** | ***p*-Value** | **% (*n*)** | ***p*-Value** | **% (*n*)** | ***p*-Value** |
| *BRCA1* PV | 50.0% (12) | 0.181 | 52.2% (12) | 0.259 | 41.7% (20) | 0.634 | 75.6% (34) | 0.985 |
| *BRCA2* PV | 66.0% (35) | | 66.0% (33) | | 37.5% (33) | | 75.7% (53) | |
| Proband | 55.6% (15) | 0.510 | 59.3% (16) | 0.803 | 32.5% (13) | 0.276 | 79.4% (27) | 0.612 |
| Personal history of BC | 54.5% (6) | 0.633 | 45.5% (5) | 0.231 | 64.7% (22) | **<0.001** | 81.4% (35) | 0.267 |
| 1st degree relative(s) with BC | 54.5% (18) | 0.264 | 51.6% (16) | 0.159 | 47.7% (31) | 0.082 | 78.9% (45) | 0.887 |
| 1st degree relative(s) with OC | 56.3% (9) | 0.628 | 68.8% (11) | 0.444 | 32.0% (8) | 0.331 | 71.4% (15) | 0.381 |
| Assessed by medical oncology | 71.4% (30) | **0.041** | 56.1% (23) | 0.270 | 41.9% (31) | 0.590 | 86.4% (57) | **0.003** |
| Assessed by ICPC | 63.3% (31) | 0.596 | 63.0% (29) | 0.748 | 38.8% (31) | 0.750 | 83.9% (52) | **0.038** |
| Received specialty care ‡ | 66.1% (39) | 0.099 | 60.7% (34) | 0.767 | 42.3% (44) | 0.273 | 84.1% (74) | **<0.001** |
| Primary care provider on record | 65.2% (43) | 0.539 | 59.4% (38) | 0.105 | 42.4% (50) | 0.662 | 79.4% (81) | 0.092 |
| Rural residence § | 55.9% (19) | 0.409 | 65.6% (21) | 0.537 | 35.7% (20) | 0.515 | 72.9% (35) | 0.563 |
| Distance to MRI > 2 h | 50.0% (4) | 0.499 | 62.5% (5) | 0.958 | 57.9% (11) | 0.068 | 75.0% (12) | 0.948 |
| Distance to surgical center > 2 h | 37.5% (3) | 0.149 | 62.5% (5) | 0.958 | 55.6% (10) | 0.121 | 75.0% (12) | 0.948 |
| Age ≥ 50 years | 65.0% (26) | 0.459 | 62.5% (25) | 0.868 | 45.1% (37) | 0.070 | 75.9% (60) | 0.912 |

\* Percentages of eligible women compliant with each intervention per factor category are shown; total numbers are shown in brackets. † Significant $p$-values (<0.05) are shown in bold font. ‡ Includes those women who have been assessed by a medical oncologist and/or the ICPC. § Rural areas are those with postal codes containing zero as the first number (e.g., A0A 1A3).

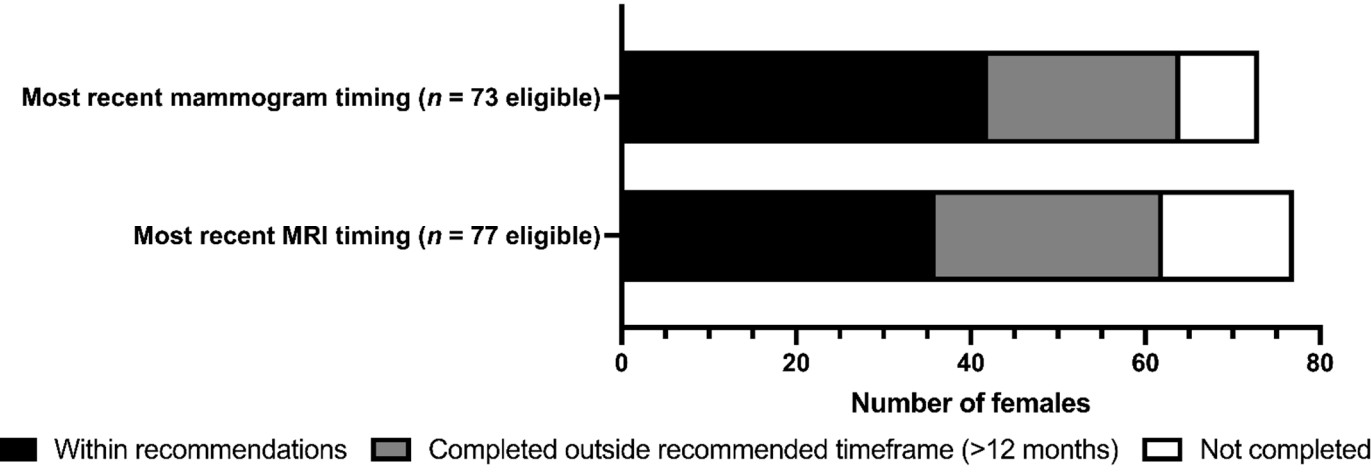

**Figure 3.** Uptake of breast MRI and mammographic screening in female NL *BRCA1/2* PV carriers.

When cases were categorized by three levels of adherence to recommendations, multi-nomial logistic regression identified that access to specialty care was the most important factor influencing compliance with optimal screening and prevention (Table 4). Females who had received specialty care were more likely to be very adherent to prevention or screening (73.2% versus 13.4%; odds ratio (95% confidence interval) = 0.249 (0.096–0.647);

*p* = 0.004). The presence of a family physician on record, urban home community, family cancer history, and older age were also associated with higher compliance.

**Table 4.** Multinomial logistic regression analysis of factors influencing screening and risk-reducing intervention adherence level among eligible female NL *BRCA1/2* PV carriers.

| Factor | Very Adherent * | Adherent | OR (CI) † | *p*-Value †‡ | Non-Adherent | OR (CI) | *p*-Value |
|---|---|---|---|---|---|---|---|
| *BRCA1* PV | 66.7% (36) | 7.4% (4) | 1.581 (0.685–3.651) | 0.122 | 25.9% (14) | 0.399 (0.124–1.278) | 0.283 |
| *BRCA2* PV | 65.6% (61) | 18.3% (17) | | | 16.1% (15) | | |
| Personal history of BC and/or OC | 74.0% (37) | 12.0% (6) | 0.649 (0.231–1.820) | 0.411 | 14.0% (7) | 0.516 (0.201–1.326) | 0.170 |
| Personal history of other cancer(s) | 60.0% (6) | 30.0% (3) | 2.528 (0.578–11.052) | 0.218 | 10.0% (1) | 0.542 (0.063–4.692) | 0.578 |
| 1st degree relative(s) with BC | 68.1% (47) | 14.5% (10) | 1.307 (0.457–3.738) | 0.617 | 17.4% (12) | 0.998 (0.399–2.496) | 0.997 |
| 1st degree relative(s) with OC | 56.0% (14) | 28.0% (7) | 3.800 (1.238–11.664) | **0.020** | 16.0% (4) | 1.143 (0.338–3.870) | 0.830 |
| Assessed by medical oncology | 76.3% (61) | 12.5% (10) | 0.537 (0.207–1.388) | 0.199 | 11.3% (9) | 0.312 (0.126–0.774) | **0.012** |
| Assessed by ICPC | 69.9% (58) | 15.7% (13) | 1.093 (0.414–2.882) | 0.858 | 14.5% (12) | 0.576 (0.241–1.378) | 0.215 |
| Received specialty care § | 73.2% (82) | 13.4% (15) | 0.457 (0.153–1.367) | 0.161 | 13.4% (15) | 0.249 (0.096–0.647) | **0.004** |
| Primary care provider on record | 72.1% (93) | 11.6% (15) | 0.108 (0.017–0.698) | **0.019** | 16.3% (21) | 0.452 (0.039–5.216) | 0.524 |
| Rural residence | 56.9% (33) | 22.4% (13) | 3.152 (1.188–8.362) | **0.021** | 20.7% (12) | 1.369 (0.585–3.203) | 0.469 |
| Distance to MRI > 2 h | 68.4% (13) | 15.8% (3) | 1.077 (0.278–4.174) | 0.915 | 15.8% (3) | 0.746 (0.197–2.820) | 0.665 |
| Distance to surgical center > 2 h | 57.9% (11) | 21.1% (4) | 1.840 (0.523–6.466) | 0.342 | 21.1% (4) | 1.2551 (0.366–4.271) | 0.721 |
| Age ≥ 50 y | 66.7% (60) | 21.1% (19) | 5.858 (1.290–26.613) | **0.022** | 12.2% (11) | 0.377 (0.160–0.866) | **0.025** |

CI: 95% confidence interval; OR: odds ratio. * Percentages of eligible women per adherence and factor category are shown; total numbers are shown in brackets. † Very adherent versus adherent. ‡ Significant *p*-values (<0.05) are shown in bold font. Very adherent versus non-adherent. § Includes those women who have been assessed by a medical oncologist and/or the ICPC.

## 4. Discussion

This retrospective review reports a comprehensive population-based dataset of 156 females at very high cancer risk due to a *BRCA1/2* PV. We observed that a substantial number (39%) of eligible females had not accessed breast MRI and/or mammographic surveillance in accordance with recommended guidelines, and that only 66% of all cases who were eligible for both breast screening and BC/OC risk reduction were fully adherent to recommendations. Specialized cancer genetics clinic care was strongly associated with successful adherence to screening and surgical prevention.

BC was more common and occurred at younger ages in *BRCA1* PV carriers compared to *BRCA2* PV carriers. These findings are supported by the literature, as the peak incidence rate for *BRCA1* PV carriers for BC and OC is 5 to 10 years earlier than for those with *BRCA2* [2]. Breast screening uptake among eligible females was lower for MRI (61.0%) and mammogram (61.6%) compared to the published Canadian rates of 76.7% and 96.5% for each modality, respectively [32]. Adherence to both MRI and mammogram within an 18-month timeframe was only 41.6%, which was lower than the 49% compliance rate reported using a 15-month surveillance period in a US report [33]. In our clinic, patients often report difficulties booking MRI, with barriers related to scheduling around menstrual cycles, limited numbers of imaging appointments, and travel. Our team is currently exploring patient-reported experiences to characterize barriers to access.

The RRM rate of 39.0% is consistent with reported Canadian rates (38.0%–41.2%) [32,34]. RRM uptake was significantly higher in females with a BC diagnosis preceding genetic

testing compared to those unaffected at the time of genetic testing; this association has been reported for RRM uptake in Canada [34] and elsewhere [35,36]. The variation in RRM uptake based on prior BC may be due to the proportion of females who had already had unilateral mastectomy. We have previously published on NL patient decision making around mastectomy; many patients report that a prior BC reduces their tolerance for future risk of a subsequent cancer and treatment, and cosmetic concerns about breast symmetry may prompt females to pursue subsequent contralateral mastectomy [37]. A significant majority (57.7%; *p* < 0.001) of the females in this study who underwent RRM elected to have reconstruction.

The rate of RRSO for NL *BRCA1/2* PV carrier females was 75.7%, higher than reported by other clinics worldwide (36.7–71.8%) [32], and is attributed to specialist assessment, strengthened by a dedicated cancer genetics clinic run by the gynecologic oncology team, close relationships within the medical community, personalized menopause care, and outreach and educational programming led by local OC advocacy groups. Specialist counselling has been associated with higher RRSO uptake in high-risk females in other jurisdictions [38,39].

The frequency of *BRCA1* and *BRCA2* PVs observed in this cohort and their recurrence among multiple families is consistent with NL's known ancestry [28] and suggests evidence for multiple founder effects [40]. Other founder effects predisposing to hereditary cancers in the NL population have been described [41–43]. As NL has the highest and second highest incidence rates of female BC and OC in Canada, respectively [44], identifying and understanding the contributions of these *BRCA1/2* PVs to hereditary BC and OC in NL is critical. *BRCA2* PVs were observed more commonly than *BRCA1* PVs (63.5% versus 36.5%), which differs from other Canadian jurisdictions reporting higher rates of *BRCA1* PVs [34,45–47]. The higher proportion of *BRCA2* PVs in NL is attributed to the presence of several large, multigenerational *BRCA2* pedigrees. Overall, the number of NL *BRCA1/2* PV carriers identified to date is lower than expected based on a reported prevalence of approximately 0.7% in a population with Northern European Caucasian ancestry [48,49]. Only 276 *BRCA1/2* PV carriers have been identified since the advent of clinical genetic testing in NL up to the date of this project, which is 0.05% of the province's 510,550 residents [27]. This is likely due to health authority policy at the time, which permitted publicly funded *BRCA1/2* testing only for individuals with a cancer diagnosis or known familial PV. The true *BRCA1/2* PV prevalence rate in an unselected NL population remains unknown; if estimated at approximately 1:137, then a population-based ascertainment regardless of cancer diagnosis would be expected to identify >3500 individuals rather than the 276 observed.

The mean age at genetic testing was 47.1 years in this study, with *BRCA1* PV carriers tested at a significantly younger age than those with a *BRCA2* PV (43.7 years versus 49.1 years). These findings are in keeping with other Canadian publications reporting genetic testing between ages 45.6 years and 49.1 years [34,45]. Most females were, therefore, identified more than 10 years later than the age at which evidence supports initiation of screening and prevention. Indeed, 31.4% of *BRCA1/2* PV carriers developed cancer prior to genetic testing, a substantial proportion of which may have been detected earlier through breast surveillance or prevented completely with surgery had these healthy individuals been aware of their high-risk status. Many authors argue that the identification of a *BRCA1/2* PV in a person after cancer diagnosis is a failure of prevention [45]. If a *BRCA1/2* PV is identified in a cancer patient, unaffected relatives may then be offered testing and access to prevention and screening. In other words, in the current model of care, at least one person must develop cancer before there is any potential to prevent another.

In Canada, publicly funded genetic testing to identify people at risk of cancer predisposition syndrome is available to those with a cancer diagnosis or to unaffected people with either a known familial mutation or a pretesting estimated family history risk >5–10%. [50]. This strategy was created during a time when very few genetic tests were available, testing costs were high, and the standard of care required individual pre- and post-test genetic counselling. The current structure of cancer prevention care for high-risk individuals in

Canada is both the most expensive and the least effective. Our team has a clear conclusion: opportunities for prevention have been missed. The key observations of both this study and similar projects are (1) many high-risk females in Canada do not receive the dedicated specialty care required to prevent cancers and improve outcomes, and (2) the current family-history-based testing strategy to identify females with a *BRCA1/2* PV misses a significant proportion of those at very high cancer risk. Health policies that focus on broader strategies to identify those at high cancer risk are needed. Given the current affordability of genetic testing and the clear evidence that preventative interventions are highly effective, Canadian health systems should direct resources towards an outcome-driven hereditary cancer care policy that ensures that all Canadians at high cancer risk receive optimal prevention.

*Strengths and Limitations*

Many studies that evaluate intervention uptake in high-risk females rely on patient questionnaires or patient visits at a single center and are, therefore, vulnerable to recall bias and incomplete data. The data generated here resulted from comprehensive reviews of each patient's clinical chart in the setting of an electronic medical record, a population-based healthcare system, a single cancer care program, and a province-wide genetics service. This centralized system allowed for a complete dataset compilation, capturing uptake of all medical interventions. Each individual case was assessed for eligibility for each specific intervention. For example, females with one breast were still considered eligible for breast surveillance and contralateral RRM, and those below age 30 were considered eligible for MRI but not breast mammography according to recommendations. This dataset, therefore, represents an accurate assessment of the real utilization of screening and prevention interventions. Given that this is a true population-based study, bias regarding referrals as a predictor of attendance at high-risk clinics has been excluded.

Despite the study strengths, there are limitations. In rare cases, pathology reports of historical cancer cases were not available. Two carriers were found to have a *BRCA1/2* PV through private genetic testing, but it is possible that other privately ascertained NL *BRCA1/2* PV carriers are missing. Direct to consumer genetic testing was available only towards the latter years of this project, making it less likely that many private testing cases were missed. We did not exclude patients with OC from eligibility for breast screening, a potential criticism of the study, as some of these cases may have advanced or incurable disease. The decision to offer breast screening to patients with OC should be considered on a case-by-case basis, but given the low absolute numbers of OC, this is not likely to influence the conclusions.

Predictors of adherence reported in previous studies were not available for exploration here, including parity, endometriosis, ethnicity, higher income, a higher knowledge and awareness of the topic, and elevated perception of cancer risk and associated anxiety [33,36,51]. Thus, the impact of these influences on the intervention behavior of NL *BRCA1/2* PV carrier females remains to be determined. Although the important role of the patient experience is not addressed within this manuscript, patient partners are members of our research team and were included throughout the study's design, execution, and analysis. Qualitative and quantitative patient-oriented projects informed by our patient partners are ongoing. These studies are exploring patient-specific barriers to successful adherence, such as the role of psychological barriers to interventions using surveys, validated scales, and qualitative interviews.

It can be argued that the term "adherence" can be paternalistic or may not completely capture all of the elements influencing an individual patient's uptake of specific interventions or conformity to guideline recommendations. Our team was very thorough in the assessment of each case to determine the exact interventions that would have been appropriate for each individual, however, it is possible that other factors could be at play in personalized patient care. For example, a patient with significant co-morbidities that substantially increased surgical risk may be counselled against RRSO. We are aware of one

patient in this dataset in that circumstance; specifically, a patient with severe cardiac disease for whom the surgical risks were determined to outweigh the potential benefit of RRSO.

Given that it is now understood that high-grade serous cancers arise in the fallopian tube, many centers will consider the option of a two-step procedure for some patients, offering salpingectomy alone and later completion of oophorectomy. This is not yet standard of care because the true extent of the cancer risk reduction of salpingectomy alone in *BRCA1/2* PV carriers is not yet known. Several large international trials addressing this question are in progress, and data may be expected in several years. At the time of this project it was not routine to offer a two-step procedure to NL *BRCA1/2* PV carriers. In the latter years of the study period, salpingectomy alone with sectioning and extensively examining the fimbriated end (SEE-FIM) pathologic processing of the fallopian tubes could have been offered in select cases when a patient was (a) younger than the recommended age of RRSO or (b) requesting permanent surgical sterilization for contraception. We are not aware of any individuals in this dataset who availed of this option during the study timeline.

Although exploration of the use of olaparib or other poly-ADP ribose polymerase (PARP) inhibitors in the care of *BRCA1/2* PV carriers is beyond the scope of this project, we note that these agents are currently a key element of pharmacologic treatment for patients with advanced high-grade serous tubo-epithelial carcinoma or metastatic BC and confer improved progression-free survival rates [52]. Data on theutilization of these agents was not collected in this study. We believe that this does not affect the conclusions, as these prescriptions would not influence any recommendations about screening or prevention for other cancer primaries, and it is not yet known if PARP inhibitor use for one type of cancer influences the occurrence of other cancers. Studies addressing this question are needed.

This study has demonstrated that access to specialty care, especially attendance at a hereditary cancer prevention clinic, is one of the most important predictors of intervention uptake. It would be helpful to study the underlying process by which some women accessed specialty clinics whilst others did not. The current model in our center relies on opportunistic referral from either physicians or the genetics department without a formalized registry. The development of programmatic processes by which all *BRCA1/2* PV carriers are seen by a dedicated service will be a key element of any improved care model. It is possible that patients who are less motivated to undertake optimal prevention may choose not to attend the high-risk clinic. Perhaps specialty care could be less of a predictor of uptake than underlying patient personality and preference. Our team is currently conducting a study exploring patient-reported experiences to characterize barriers to access, including patient understanding and tolerance of risk, anxiety, and avoidance of health seeking behaviors.

## 5. Conclusions

The care of Canadian *BRCA1/2* PV carriers represents a disconnect between evidence and health policy. These data reveal that consultation with specialty care is the strongest predictor of adherence to breast surveillance and uptake of RRSO. Those patients who were not engaged with a specialty care team were less likely to follow guideline-based recommendations. Access to cancer genetics expertise should be a routine component of the ongoing care of these patients and their families. A Canadian Hereditary Cancer Registry would include several elements, including patient support in navigation of surveillance and surgeries, assistance with at-risk relative recruitment, patient education, family physician outreach and support, and ongoing quality assurance. All patients should be offered opportunities to participate in research. Further exploration of patient-reported experiences to characterize barriers to access will also be valuable in determining if this study's results could also reflect patients' personalities and preferences rather than system deficiencies. A future Canadian high-risk cancer prevention strategy will require a programmatic, patient-centered model and proper evaluation of both individual and health system factors influencing successful cancer prevention.

**Author Contributions:** A.R. reviewed all individual cases in detail and drafted the first and subsequent edits of the manuscript. H.E. and M.S. were engaged in this project from the outset, including grant application, study design, and manuscript editing. K.N.S. carried out data analysis and substantial manuscript edits. L.D. designed the original concept and oversaw the study, the engagement with patient partners, the analysis, and manuscript drafts and edits. All authors have read and agreed to the published version of the manuscript.

**Funding:** This work is made possible by competitive funding through the NL SUPPORT Patient-Oriented Research Grant number 210399, St. John's, NL.

**Institutional Review Board Statement:** Ethical approval was granted by the Human Research Ethics Authority of NL (protocol#14.054) and the Research Proposal Approval Committee (Eastern Health Authority). The study was conducted in accordance with the Declaration of Helsinki.

**Informed Consent Statement:** This project involved clinical case review only without direct patient intervention. Patient consent was waived as per the research ethics board's approval.

**Data Availability Statement:** Data available on request due to restrictions eg privacy or ethics. The data presented in this study are available on request from the corresponding author. The data are not publicly available due to protection of privacy or participants.

**Acknowledgments:** We are grateful to Margaret Steele, Dean, Faculty of Medicine for her encouragement of the NL Ovarian Cancer Research Program, and to Kimberley Manning RN and medical student Ashely Gabriel for their work on this project. We thank Sharon Smith RN and Nancy Wadden MD for their advice. The work of this team is possible only through the enthusiastic support of the "Belles with Balls" Ovarian Cancer Education and Research Team, St. John's, NL. We are always in service to the women of NL, and especially thank patient partners Diane Farrell, Jamie Farrell, Cindy Cranford, Lenora Walsh and Angelina Hayes. Your advice on this work was invaluable.

**Conflicts of Interest:** The authors declare no conflict of interest. The funders had no role in the design of the study, in the collection, analyses, or interpretation of data, in the writing of the manuscript, or in the decision to publish the results.

## Abbreviations

BC: breast cancer; HRT: hormone replacement therapy; ICPC: Inherited Cancer Prevention Clinic; MRI: magnetic resonance imaging; OC: ovarian cancer; PV: pathogenic variant; RRM: risk-reducing mastectomy; RRSO: risk-reducing salpingo oophorectomy; SEE-FIM: sectioning and extensively examining the fimbriated end.

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
