# Peer review of "Specialty Care and Counselling about Hereditary Cancer Risk Improves Adherence to Cancer Screening and Prevention in Newfoundland and Labrador Patients with BRCA1/2 Pathogenic Variants: A Population-Based Retrospective Cohort Study"

_curroncol, doi:10.3390/curroncol30100678_

Round 1

Reviewer 1 Report

I read the article “Dedicated care improves adherence to cancer screening and prevention in Newfoundland women with BRCA pathogenic variants: a population-based retrospective cohort study” with great interest. The study deals with a very important topic: how to translate and communicate the recommendations of the guidelines for women with a pathogenic variant in BRCA in such a way that the highest possible adherence is achieved. The study is based on retrospective survey and is carefully conducted and described throughout. My main criticisms relate to possible misunderstandings that might be generated. I list my main points along the structure of the manuscript.

Already the title could create a misunderstanding. When I first read it, I thought that "dedicated care" meant commitment, in the sense of particularly supportive care. What is meant, however, is specialised counselling.

Introduction: The second major point, which already arises in the abstract, refers to the concept of adherence. Is it possible to characterise adherence based on statistics alone? How does one know what was recommended to the patient in an individual case? Of course, there is a certain probability that the guideline recommendations should be consistent with the patient's implementation, but this is not entirely certain. Therefore, I would perhaps rather use the term "conformity with guideline recommendations". Otherwise, a rather paternalistic picture might painted, which is certainly not the intention of the authors.

The cancer prevalence for women with a pathogenic variant provides the reader with a range (e.g. 51-72%), but the same is not done for the numbers of the general population. Maybe limit the reference for the BRCA carriers as well?

The research question could be better derived and substantiated with references in the introduction, e.g. the importance of specialised counselling and other factors that will be examined later.

Method: ≥18 years or >18 years?

Results: Is testing done only when there is a known mutation in the family present, or also if there is a family history of cancer?

Line 12/13 from above: please delete any interpretation in the results section in the results section.

Table 3 refers to the regression analysis. Please mention the regression analysis in the heading if true. The parameters of a rgression analysis (e.g. OR and CI) that would allow an interpretation, are missing here. On the basis of the significance alone, it is not possible to determine the strength of the influence (e.g. specialty care, p = 0.004). In this respect, the conclusion is not really comprehensible on the basis of the data.

Discussion: The discussion relates to important points and puts the results in a context with international studies. At the end of the discussion the authors point to the patients' experience and preferences and mention that the patients should be taken into account.

Conclusion: here the confusion of "specialised" with "dedicated" occurs again.

Author Response

Thank you to the reviewers for these thoughtful comments. We have drafted a summary letter reviewing each reviewer's points and addressing the relevant changes or edits.

  1. The comment that the term “dedicated care” could create misunderstanding is a good point. We have changed this to “Speciality care and counselling for hereditary cancer prevention”.
  2. It is quite true that the term “adherence” can be seen as paternalistic or judging of those patients who do not have medical care in line with recommendations. This point has been expanded in the discussion. Our team was very thorough in the assessment of each case to determine the exact interventions that would have been appropriate for each individual, however, the reviewer is correct to state that other factors could be at play in personalized patient care. For example, a patient with significant co-morbidities that substantially increased surgical risk may be counselled against risk-reducing salpingectomy (RRSO). We are aware of one woman in this dataset in that circumstance, a patient with severe cardiac disease for whom surgical risks were determined to outweigh the potential benefit of RRSO.
  3. The risks of breast and ovarian cancer in the general population are included in the opening paragraph of the manuscript to contrast with the large elevation of risk for BRCA1/2 PV carriers.
  4. The core point of this research question, to determine which factors most influence conformity with guideline recommendations and how these results can assist development of future program development, has been expanded.
  5. The dataset included patients at or above 18 years of age; this has been clarified.
  6. Genetic testing in this jurisdiction is offered when there is a known familial mutation or when a patient with cancer meets eligibility testing criteria with the NL Provincial Medical Genetics Program.
  7. Interpretation comments have been removed from the results section and re-organized in the discussion.
  8. Odds ratios and 95% confidence intervals have been added to the multinomial logistic regression table, and reference to the regression analysis has been added to the table heading.
  9. The term dedicated has been removed and the phrasing refocused to be clear that we mean speciality care.

Lesa Dawson MD FRCSC

Reviewer 2 Report

This an excellent manuscript presenting critically important information.  Well-designed, well-written and with appropriate and meaningful conclusions/recommendations.

My only suggestion is to consider adding a discussion of risk-reduction bilateral salpingectomy vs. bilateral salpingo-oophorectomy.

Author Response

Thank you for your positive comments on this work.

The point about the importance of mentioning risk-reducing salpingectomy vs. risk-reducing salpingo-oophorectomy is an important one. Thank you for raising this. We have inserted a comment about this point in the discussion. Given that there is no data yet confirming the extent of cancer risk reduction with salpingectomy alone in BRCA carriers, we cannot yet state that this intervention is the standard of care. At the time of this project, it was not routine to offer a two-step procedure to carriers. In the latter years of the study period, salpingectomy alone with SEE FIM protocol pathologic processing of the fallopian tubes may have been offered in select cases when a patient was a) younger than the recommended age of RRSO or b) requesting permanent surgical sterilization for contraception. We are not aware of any individuals in this dataset who availed of this option during the study timeline.

Lesa Dawson 

Reviewer 3 Report

The above MS presents a population-based retrospective cohort study that analyses the adherence to cancer screening and prevention in Newfoundland women with BRCA pathogenic variants (PV).

This study has demonstrated that access to specialty care, especially attendance at a hereditary cancer prevention clinic is one of the most important predictors of intervention uptake.

Comments to Authors have been appended below.

In the Introduction [p. # 1]

Pathogenic variants (PV) in these genes cause hereditary cancer predisposition syndrome (CPS) resulting in lifetime risks of breast (BC) and ovarian (OC) cancer of 51-72% and 11- 44% respectively, a significant contrast when compared to rates of 13% and 1.3% in the general population [2,3,4].

please, remove the 2-nd/redundant: ‘of 13% and 1.3%.’[at the end of this sentence]

In the Discussion [p. # 9], the Authors may briefly comment on possible pharmaco-therapeutic implications of having BRCA1 or BRCA2 PV for patients who have already developed BC. Since BRCA1 and BRCA2 genes are involved in DNA repair, tumors with such alterations are susceptible to some anticancer therapies, which damage DNA (e.g., PARP inhibitors, including olaparib or talazoparib, and chemotherapeutic agent cisplatin).

Also, patient input could be essential for future designs of research studies and educational programs. Perhaps, the Authors could address the main directions in investigating the patient’s perspective on ‘barriers to access specialty care clinics’. Similarly, it might be beneficial to explore the patient’s psychological profile, to understand better her motivations, expectations, fears, and needs, as well as encourage participation in cancer prevention, in a more personalized way. Practical education/support for patients and clear communication patterns with medical team members must be implemented to achieve necessary clinical improvements in this area.

In the Conclusions [p. # 9], the Authors may briefly elaborate on ‘A future Canadian high-risk cancer prevention strategy ...’ In particular, it should be emphasized that the risk-reducing surgery involves bilateral mastectomy to significantly decrease the risk of BC, and bilateral salpingo-oophorectomy to decrease the risk of OC. Moreover, it needs to be underscored that in premenopausal women, removing the ovaries may also decrease the risk of BC by eliminating a source of female hormones (which can promote the growth of hormone-sensitive BC subtypes). Risks and possible adverse effects (AE) of the prophylactic procedures need to be clearly explained to the patients to enable them to make the most optimal, informed decisions.

A list of abbreviations will be useful [at the end].

Author Response

Thank you for your helpful comments and edits.

  1. The recommended edits to the opening sentence about the cancer in the general population have been made.
  2. It is correct to state that the role of Olaparib and other PARP inhibitors is very important in the treatment of both high-grade serous tubo-ovarian cancer and metastatic breast cancer in BRCA1/2 PV carriers and that any research work about this population should always consider how the use of these agents might influence patient care. We did not collect data on PARP utilization as the Health Canada approval was only granted towards the end of the study period. A paragraph addressing the importance of these agents in the care of BRCA-associated cancers has been added to the discussion.
  3. Thank you for the comments about the role of the patient perspective around uptake and adherence. As stated in the discussion, it may well be true that certain patients with anxiety, perceived lack of autonomy or fear of the medical system might not avail of screening or prevention even under ideal circumstances. Our team conducted a parallel project that administered surveys and validated scales about worry/genetic disease and conducted interviews with BRCA1/2 PV carriers about these questions. This is mentioned in the discussion.
  4. The respective risk reduction of RRSO and mastectomy is addressed in the introduction and we have clarified that RRSO improves all-cause mortality (not just ovarian cancer-related mortality), and that breast cancer risk in BRCA2 carriers may be reduced after RRSO. Thank you for this comment.
  5. A list of abbreviations has been added at the end of the text.

Lesa Dawson

Round 2

Reviewer 3 Report

Thank you for making all the modifications in this MS

Author Response

Thank you for pointing this out. We have edited the relevant sentence.